# Stereoscopic 3D geometric distortions analyzed from the viewer's point of view

**Zhongpai Gao** *, **Guangtao Zhai, Xiaokang Yang**

Artificial intelligence institute, Shanghai Jiao Tong University, Shanghai, China

* gaozhongpai@sjtu.edu.cn

## Abstract

Stereoscopic 3D (S3D) geometric distortions can be introduced by mismatches among image capture, display, and viewing configurations. In previous work of S3D geometric models, geometric distortions have been analyzed from a third-person perspective based on the binocular depth cue (i.e., binocular disparity). A third-person perspective is different from what the viewer sees since monocular depth cues (e.g., linear perspective, occlusion, and shadows) from different perspectives are different. However, depth perception in a 3D space involves both monocular and binocular depth cues. Geometric distortions that are solely predicted by the binocular depth cue cannot describe what a viewer really perceives. In this paper, we combine geometric models and retinal disparity models to analyze geometric distortions from the viewer's perspective where both monocular and binocular depth cues are considered. Results show that binocular and monocular depth-cue conflicts in a geometrically distorted S3D space. Moreover, user-initiated head translations averting from the optimal viewing position in conventional S3D displays can also introduce geometric distortions, which are inconsistent with our natural 3D viewing condition. The inconsistency of depth cues in a dynamic scene may be a source of visually induced motions sickness.

## 1 Introduction

The goal of display systems is to convey the real world or virtually constructed 3D worlds veridically to viewers. Compared to 2D displays, stereoscopic 3D (S3D) displays are able to provide binocular disparity depth cue. Various S3D display technology has been used for virtual/augmented reality, scientific visualization, medical imaging, 3D movies, and gaming. However, orthoscopic presentation of the 3D scene remains a challenging task. Parametric mismatches occur among stereoscopic capture, display, and viewing processes cause geometric distortions for the viewer [1–4]. Motions in geometrically distorted S3D space are suspected as a potential cause of visually induced motion sickness (VIMS) [3].

In general, VIMS is considered as a physiological response induced by inter-sensory motion signal conflicts, e.g., the motion signal conflicts between the visual and vestibular systems [5, 6]. However, the inter-sensory conflict theory cannot explain why watching S3D videos causes significantly higher levels of discomfort [7] and motion sickness [8] than watching 2D videos. Another explanation of VIMS is sensory rearrangement [9]—'Whenever the central nervous

**Data Availability Statement:** All relevant data are within the paper.

**Funding:** This work was supported by the National Natural Science Foundation of China (61901259) and China Postdoctoral Science Foundation

(BX2019208). The funders had no role in study design, data collection and analysis, decision to publish, or preparation of the manuscript.

**Competing interests:** The authors have declared that no competing interests exist.

system receives sensory information concerning the orientation and movement of the body which is unexpected or unfamiliar in the context of motor intentions and previous sensory-motor experience—and this condition occurs for long enough—motion sickness typically results' [10]. Hwang and Peli [3] and Gao *et al*. [4] pointed out that depth-cue conflicts in a geometrically distorted S3D space with motions may cause VIMS, which can be explained by the sensory rearrangement theory since depth-cue conflicts are unexpected in the real 3D world motion.

Depth perception in a 3D space involves monocular and binocular depth cues. Monocular depth cues include static monocular depth cues, also called pictorial depth cues [11], and motion parallax [12]. Pictorial depth cues include linear perspective, interposition (occlusion), object sizes, shades and shadows, texture gradients, accommodation and blur, aerial perspective, etc. Motion parallax is the relative movement of images across the retina resulting from the movement of the observer or the translation of objects across the viewer's field of view. Binocular depth cues come from two space-separated eyes, including convergence and binocular disparity. In the real world, different depth cues are consistent. Human visual systems interpret depth by integrating various depth cues [13–15].

However, depth-cue conflicts occur in many situations. The most typical example is pseudoscope, which was originally invented by Wheatstone [16]. The pseudoscope optical device reverses the relationship between physical depth and binocular disparity by presenting the left eye view image to the right eye and the right eye view to the left eye, vice versa. Therefore, the device provides a scene with binocular depth reversed, while monocular depth cues are veridically preserved. Consequently, monocular and binocular depth cues conflict (reversed), which may cause sickness. Depth-cue conflicts can also occur in S3D viewing with geometric distortions.

Several geometric models have been built to predict geometric distortions in S3D. Woods *et al*. [2] proposed a transfer function from the real (or virtual) world to the S3D world. Using this model, various geometric distortions, such as depth plane curvature (i.e., objects are bent away from the viewer in the periphery), depth non-linearity (i.e., depth differences in the reconstructed world do not match the corresponding depth differences in the original world), and shearing distortion (i.e., objects appear sheared toward the viewer's head position), were discussed. Masaoka *et al*. [17] and Yamanoue *et al*. [18] built geometric models to predict two abnormal perceptions: the puppet-theater effect [19] and the cardboard effect [20].

Gao *et al*. [4] provided a geometric model and illustrated various S3D distortions caused by four parameter-pair mismatches during the image capture, display, and viewing processes: 1) camera separation vs. eye separation, 2) camera field of view (FOV) vs. screen FOV, 3) camera convergence distance vs. screen distance, and 4) head position vs. display position. In this model, the impact of each paired parameters on S3D geometric distortions was analyzed independently. The model also provided methods to correct the geometric distortions by individually matching the parameter and combining the distortion patterns to compensate for each other so that the overall distortion can be minimized.

The geometric models [2, 4, 17, 18] predict geometric distortions by the ray-intersection method [21, 22] that calculates the intersection of two projection lines from the left and the right eye to the corresponding left and right onscreen points. Thus, only the binocular disparity depth cue is considered in these geometric models. However, human visual systems interpret depth by integrating both monocular and binocular depth cues [13–15]. Perceived depth in complex S3D scenes often differs from these geometric predictions based on binocular disparity alone [23]. The geometric models that demonstrate geometric distortions from a third-person perspective without considering monocular depth cues from the viewer's perspective cannot predict the viewer's S3D perception accurately [24–27].

Hwang and Peli [3] discussed depth perceptions of camera or object movements in S3D worlds from the viewer's perspective by defining angular disparity from the visual eccentricities of corresponding retinal position. This angular disparity indicates object depths relative to the angular disparity from the fixation point. Two types of geometric distortions depth plane curvature (caused by the mismatch between capture and display image plane) and shearing distortion (caused by the viewer's head translations) were analyzed. They proposed that the motions in a distorted virtual 3D space may cause vision-to-vision intra-sensory conflicts that result in VIMS. Hwang and Peli [28] further pointed out that S3D optic flow distortion may be a source of VIMS with the help of the geometric models in [4].

In this paper, the S3D distortions, caused by the four parameter-pair mismatches: 1) camera separation vs. eye separation, 2) camera FOV vs. screen FOV, 3) camera convergence distance vs. screen distance, and 4) head position vs. display position, are analyzed in the viewer's perspective by combining the geometric model proposed by Gao *et al.* [4] and retinal disparity model used in Hwang and Peli [3]. The retinal disparity model disentangles the binocular depth cue and monocular depth cues so that geometric distortions in terms of the monocular and the binocular can be analyzed separately.

## 2 S3D geometric and retinal disparity models

S3D geometric models predict geometric distortions only considering the binocular disparity. The retinal disparity model reconstructs the presented S3D scene based on the corresponding retinal projection on the viewer. Therefore, Combining the geometric model and retinal disparity model allows analyzing both linear perspective (monocular depth cue) and disparity (binocular depth cue) simultaneously.

### 2.1 S3D geometric model

In S3D, when the original world is captured by parallel-axis with shifted sensor technique (or rendered with asymmetric converging frustums), then displayed on a flat real/virtual screen, the transfer function from the original world to the reconstructed world can be expressed as

$$P = \begin{bmatrix} X_p \\ Y_p \\ Z_p \end{bmatrix} = \frac{k_s k_f k_d d_c}{Z_o(k_s - k_f k_d) + k_f k_d d_c} \begin{bmatrix} X_o \\ Y_o \\ Z_o/k_f \end{bmatrix} + \frac{k_f k_d (d_c - Z_o)}{Z_o(k_s - k_f k_d) + k_f k_d d_c} \begin{bmatrix} T_x \\ T_y \\ T_z/k_f \end{bmatrix}. \quad (1)$$

where $O = [X_o, Y_o, Z_o]^{\mathsf{T}}$ is a point in the original world, $P = [X_p, Y_p, Z_p]^{\mathsf{T}}$ is the point corresponding to $O$ in the reconstructed world in S3D, $T = [T_x, T_y, T_z]^{\mathsf{T}}$ is the offset of head position from the origin, $k_s = \frac{s_e}{s_c}$ is the ratio of eye separation to camera separation, $k_d = \frac{d_s}{d_c}$ is the ratio of screen distance to camera convergence distance, and $k_f = \frac{\tan(\alpha_{sh}/2)}{\tan(\alpha_{ch}/2)}$ is the ratio of screen FOV to camera FOV in linear scale. The transfer function is also controlled by the camera convergence distance, $d_c$. See [4] for the derivation of the geometric model.

When the paired parameters are matched, i.e., $k_s = 1$, $k_d = 1$, $k_f = 1$, and $T = [0, 0, 0]^{\mathsf{T}}$, the transfer function (1) is simplified as

$$P = [X_p, Y_p, Z_p]^{\mathsf{T}} = [X_o, Y_o, Z_o]^{\mathsf{T}} = O. \quad (2)$$

This indicates an orthoscopic displaying condition without geometric distortions.

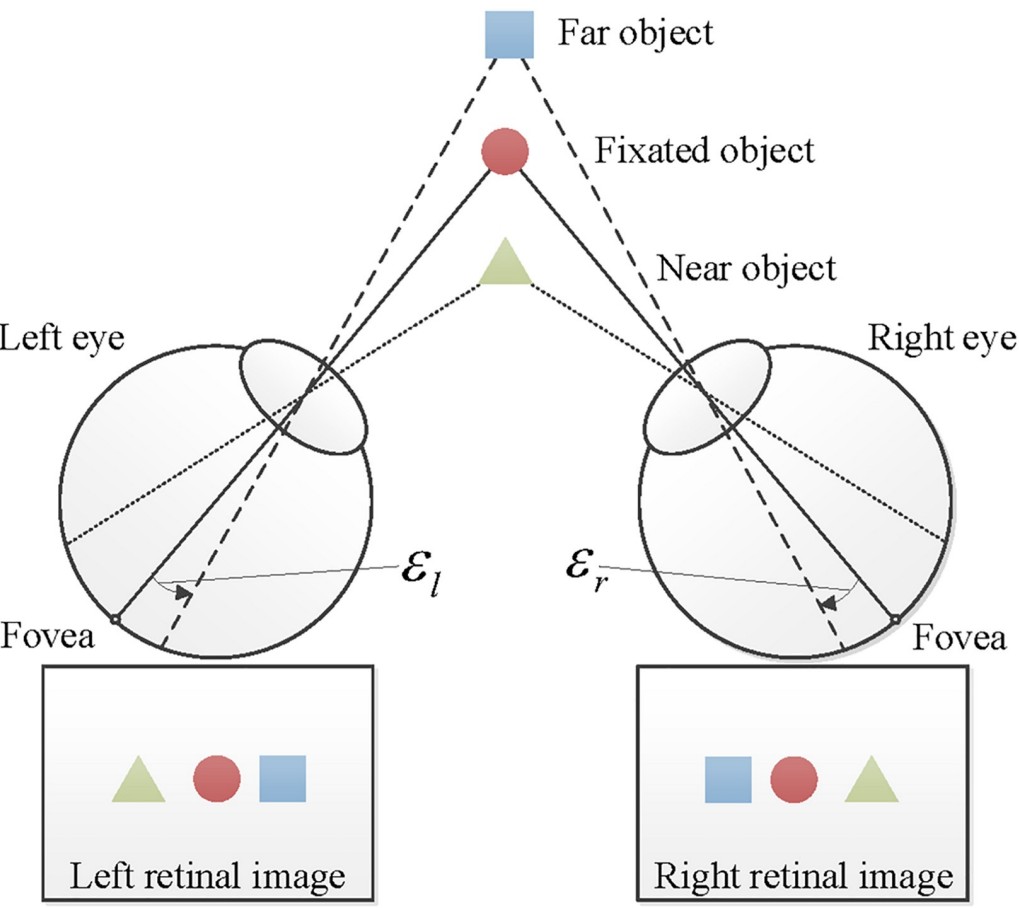

**Fig 1. Viewer perspective retinal projection of objects showing different horizontal visual eccentricities ($\mathcal{E}$), based on relative distances from the fixated point.**

## 2.2 Retinal disparity model

In the retinal disparity model [3], the object that a person fixates on is projected onto the fovea in each eye. Visual eccentricity ($\mathcal{E}$) of a point is defined as an angular distance relative to the fovea. Therefore, the eccentricity of the fixated point becomes zero ($\mathcal{E} = 0$); the visual eccentricity of a non-fixated point projected to the retina in left and right eyes are $\mathcal{E}_l$ and $\mathcal{E}_r$, respectively, as shown in Fig 1. Note that, since the geometric distortions in $x$ and $y$ dimensions are always the same (see (1)), we will only discuss $x$ and $z$ dimensions (horizontal axis and depth) in the following analyses and we only consider horizontal visual eccentricity.

The visual eccentricities of a point $A = [A_x, A_z]^\top$ (e.g., the far object in Fig 1) in the left and the right eye can be expressed as

$$\mathcal{E}_l = \quad \arctan \frac{A_x - E_{lx}}{A_z - E_{lz}} - \arctan \frac{F_x - E_{lx}}{F_z - E_{lz}}, \tag{3}$$

$$\mathcal{E}_r = \quad \arctan \frac{A_x - E_{rx}}{A_z - E_{rz}} - \arctan \frac{F_x - E_{rx}}{F_z - E_{rz}}, \tag{4}$$

respectively, where $E_l = [E_{lx}, E_{lz}]^\top$ and $E_r = [E_{rx}, E_{rz}]^\top$ are the positions of the left and the right eye, and $F = [F_x, F_z]^\top$ is the position of the fixation. Note that the visual eccentricity of a point

located at the left side of the viewer's visual field will be assigned to a negative value and that at the right visual field will be assigned to a positive value. The depth of a point relative to the fixation target can be estimated by the difference between visual eccentricities projected to the left and the right eye. This retinal angular disparity ($\mathcal{D}$) is defined as

$$\mathcal{D} = \mathcal{E}_r - \mathcal{E}_l. \tag{5}$$

Thus, the angular disparity ($\mathcal{D}$) represents the binocular depth cue.

The orientation of a point related to the viewer's perspective after binocular fusion can be defined as the visual eccentricity from the cyclopean eye, $E_c = [E_{cx}, E_{cz}]^\mathsf{T}$, where this imaginary eye is positioned midway between the left and the right eye [29, 30]. Thus, the visual eccentricity of a point to the viewer can be expressed as

$$\mathcal{E}_c = \arctan \frac{A_x - E_{cx}}{A_z - E_{cz}} - \arctan \frac{F_x - E_{cx}}{F_z - E_{cz}}. \tag{6}$$

As illustrated in Fig 2, the far and the near objects project to the same retina locations of the cyclopean eye, including all the monocular depth cues (e.g., perspective, occlusion, shade and shadow, and texture gradient). If solely based on the visual eccentricity of the cyclopean eye, the far and the near objects are indistinguishable. Thus, the visual eccentricity ($\mathcal{E}_c$) represents all the monocular depth cues.

## 2.3 A sample 3D scene structure and illustration from the viewer's perspective

Fig 3 shows the sample scene used in Hwang and Peli [3] in the real world to illustrate geometric distortions in the viewer's perspective. Objects 1–9 are arranged to be on an equally spaced rectilinear grid ($3 \times 3$ in $xz$-plane). Objects are spaced $1m$ apart in both $x$ and $z$ dimensions. The center object (O5) is the fixation object at $[0, 0, 3]^\mathsf{T}$. As an example, we consider S3D geometric distorsions when a $50mm$ IPD user watching 3D videos on a 50-inch ($1.1m \times 0.62m$) TV at $2m$ distance without lateral offset, while the scene was captured by camera separation is $63mm$, convergence distance is $3m$, and camera FOV is $45°$ (i.e., $k_s = 0.8$, $k_d = 0.67$, $k_f = 0.66$, $T = [0, 0, 0]^\mathsf{T}$).

Fig 4 shows the geometric distortions presented in eccentricity-disparity ($\mathcal{E}_c - \mathcal{D}$) coordinates. The solid and dashed lines show the original world (Fig 4a) and reconstructed world (Fig 4b), respectively. In $\mathcal{E}_c - \mathcal{D}$ coordinates, eccentricity represents monocular depth cues (i.e., perspective, occlusion, shade and shadow, and texture gradient) to the viewer. The same eccentricity of the original world and reconstructed world indicates identical monocular perception. On the other hand, disparity represents the binocular depth cue, indicating object distance relative to the fixated distance. Thus, geometric distortions from the viewer's perspective can be computed by the differences between the original world and reconstructed world in $\mathcal{E}_c - \mathcal{D}$ coordinates. The eccentricity difference and disparity difference represent geometric distortions perceived by monocular and binocular vision, respectively. The graphs are superimposed in Fig 4c to visualize the eccentricity and disparity difference.

To visually demonstrate the geometric distortions in monocular perception, the viewer's perspective (cyclopean eye) of the sample scene in Fig 3 is presented in Fig 5a. Objects 1-9 are replaced with $40cm$ brown cubes 1–9 that are arranged to be on an equally $1m$ spaced rectilinear grid ($3 \times 3$ in $xz$-plane). The fixated point is at $(0, 0, 3)[m]$. Blue cubes in Fig 5b are the reconstructed (perceived) objects in the S3D world. Blue cubes 4-6 are on the screen at $3m$ distance. Any difference between the corresponding features of the brown (Fig 5a) and blue cubes (Fig 5b) represents geometric distortions introduced by the mismatches between the

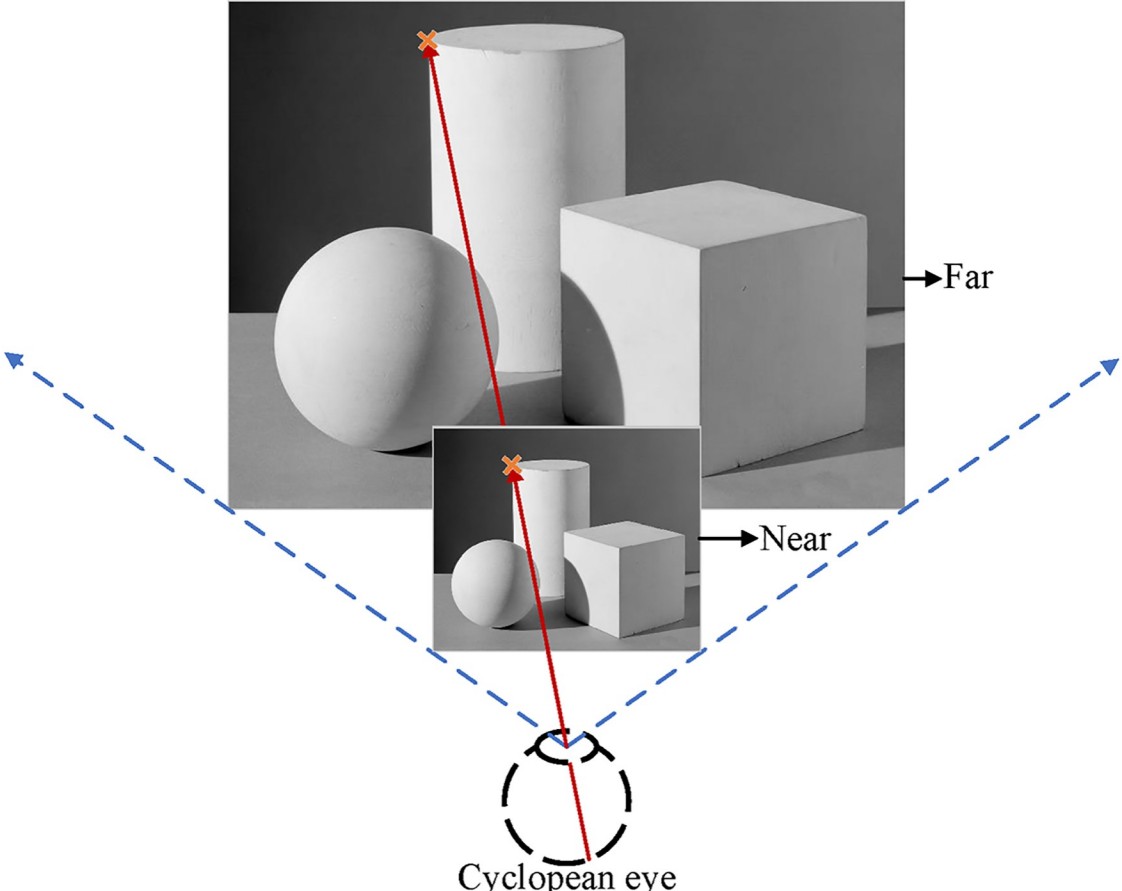

**Fig 2. Visual eccentricity from the cyclopean eye represents monocular depth cues, including perspective, occlusion, shade and shadow, and texture gradient.** The far and the near objects project to the same retina locations of the cyclopean eye and are indistinguishable based on the monocular depth cues.

capture and display in monocular perception. In subsequent simulations, the captured cubes and reconstructed cubes are superimposed on a single coordinate system to illustrate the distortions between the original world and reconstructed world, as shown in Fig 5c.

## 3 S3D geometric distortion analysis

In the following sections, we will discuss the isolated effects of parameter mismatches, assuming that the other paired parameters are matched.

### 3.1 Mismatch of camera-eye separations

This analysis assumes that screen distance and camera convergence distance are the same ($k_d = 1$), screen FOV and camera FOV are the same ($k_f = 1$), the convergence distance is constant (e.g., $d_c = 3m$), head position is at the optimal position ($T = [0, 0, 0]^\mathsf{T}$), and only camera separation and eye separation are mismatched. The transfer function in (1) can be simplified

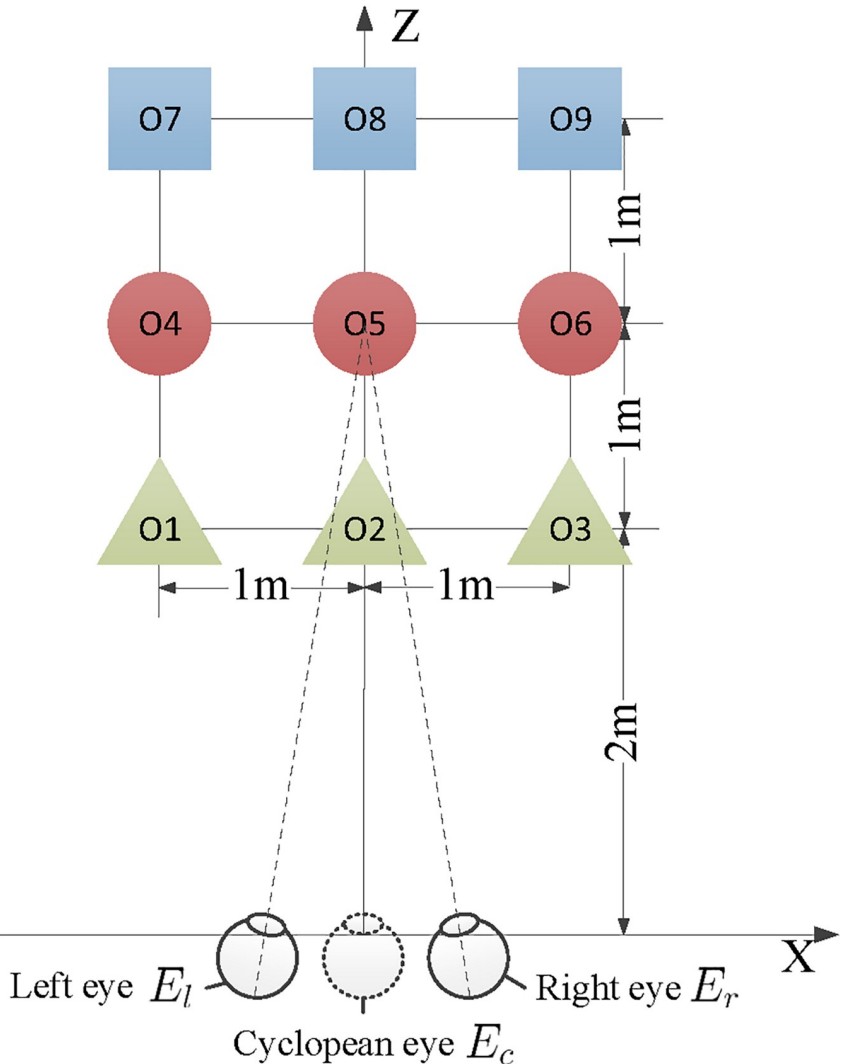

**Fig 3. Sample original scene composed of 9 objects (3-blue square, 3-red circle, and 3-green triangle) for the geometric distortion analysis.** Objects 1 to 9 are arranged to be on an equally 1m spaced rectilinear grid ($3 \times 3$ in *xz*-plane). Figure is adapted from Hwang and Peli [3] with permission.

with the ratio of eye separation to camera separation, $k_s$ as follows:

$$P = \begin{bmatrix} X_p \\ Z_p \end{bmatrix} = \frac{k_s d_c}{Z_o(k_s - 1) + d_c} \begin{bmatrix} X_o \\ Z_o \end{bmatrix}, \tag{7}$$

The visual eccentricities in the original world and reconstructed world are computed by

$$\mathcal{E}_{pc} = \arctan \frac{X_p F_{pz} - Z_p F_{px}}{X_p F_{px} + Z_p F_{pz}} = \arctan \frac{X_o F_{oz} - Z_o F_{ox}}{X_o F_{ox} + Z_o F_{oz}} = \mathcal{E}_{oc}, \tag{8}$$

by substituting $X_p$ and $Z_p$ in (7). Thus, all visual elements share exactly the same eccentricities in conditions of separation mismatches. Fig 6 shows simulations of cubes from the cyclopean eye in the original world (brown cubes) and the reconstructed world (blue cubes) when eye separation mismatches with camera separation (e.g., $s_c = 63mm$, $s_e = 50mm$, $k_s = \frac{50mm}{63mm} = 0.8$).

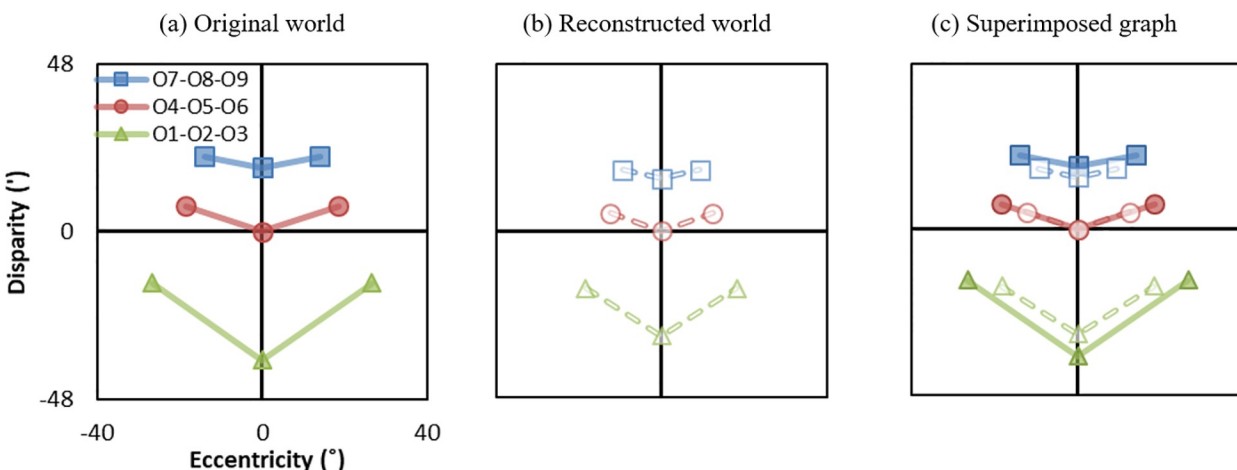

**Fig 4. Example of geometric distortions presented in eccentricity-disparity ($\mathcal{E}_c - \mathcal{D}$) coordinates.** Objects 1–9 are arranged to be on an equally spaced rectilinear grid (3 × 3 in $x - z$ plane), as shown in Fig 3. (a) Solid lines represent the eccentricity-disparity in the original world and (b) dashed lines represent the eccentricity-disparity in the reconstructed world in the condition where the separation ratio is $k_d = 0.8$, distance ratio is $k_d = 0.67$, the FOV ratio is $k_f = 0.66$, the convergence distance is $d_c = 3m$. (c) Superimposed eccentricity-disparity graph of the original world and reconstructed world. In subsequence figures, the superimposed graphs are presented to aid the visualization of distortions.

The brown cubes and blue cubes are perfectly matched. Therefore, from monocular perception, the views of the original and the reconstructed worlds are identical.

The vast majority of adults have IPDs in the $[50mm, 75mm]$ range, where the mean value of adult IPD is around $63mm$ [31], which is a recommended value of camera separation for S3D movie making [32]. Therefore, the separation ratio $k_s = \frac{s_e}{s_c}$ is in the range of $[0.8, 1.2]$.

Fig 7 shows examples with separation mismatches in $\mathcal{E}_c - \mathcal{D}$ coordinates. The solid and dashed lines represent the original world ($\mathcal{E}_{oc}, \mathcal{D}_o$) and reconstructed world ($\mathcal{E}_{pc}, \mathcal{D}_p$) when eye separation is smaller than camera separation ($k_s = \frac{50mm}{63mm} = 0.8$) and eye separation is larger than camera separation ($k_s = \frac{75mm}{63mm} = 1.2$), respectively. From binocular perception, when eye

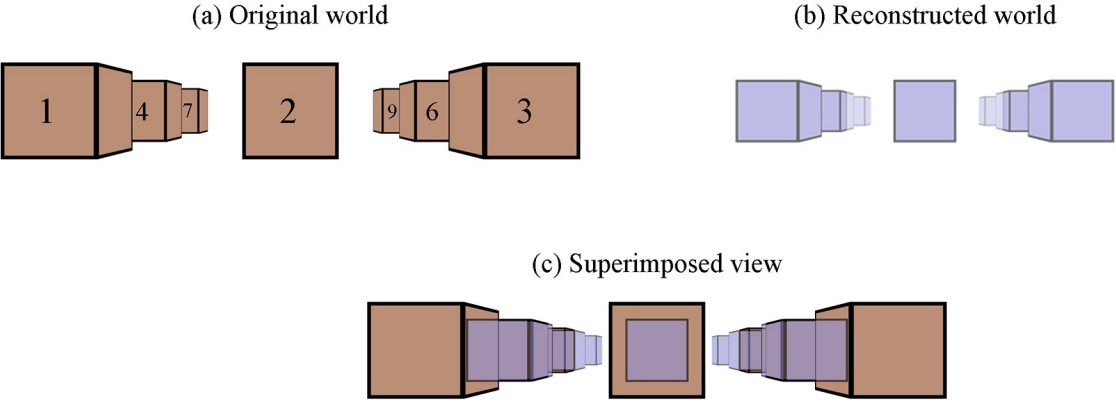

**Fig 5. The sample scene from the perspective of the cyclopean eye.** The same arrangement as in Fig 3, cubes 1–9 are arranged to be on an equally $1m$ spaced rectilinear grid (3 × 3 in $xz$ plane). The edges of the cubes are $40cm$. (a) Brown cubes are in the original world. (b) Blue cubes are the geometric distortion example (Fig 4b) in the reconstructed world corresponding to the brown cubes. The blue cubes 4-6 are on the screen at $3m$ distance. (c) Superimposed view of the original world and reconstructed world. In subsequence figures, the superimposed views are presented to aid the visualization of distortions.

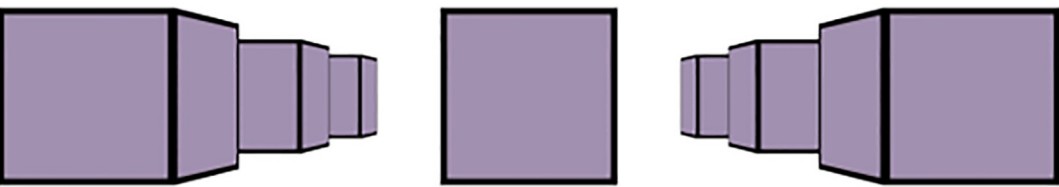

**Fig 6. Effect of separation mismatches from the cyclopean eye (i.e., only monocular depth cues).** The brown cubes are the simulations in the original world and the blue cubes are the simulations in the reconstructed world. They are identical from the view of cyclopean eye.

separation is smaller (or larger) than camera separation ($k_s = 0.8/1.2$), objects in front of the screen appear closer (or farther) and objects behind the screen appear farther (or closer). That is, objects appear expended (or compressed) in depth. Importantly, the eccentricities in the reconstructed world (dashed lines) are identical with the original world (solid lines), representing no geometric distortion from monocular perception.

A special case of separation mismatch is pseudoscope where the left view is projected to the right eye and the right view is projected to the left eye (i.e., $k_s = -1$). From the cyclopean eye's perspective, the reconstructed world is identical to the original world, as shown in Fig 6. From binocular perception (Fig 7c), onscreen objects stay on the screen, objects in front of the screen appear behind the screen, and objects behind the screen appear in front of the screen. Thus, the binocular depth cue is reversed with respect to the screen for pseudoscope.

## 3.2 Mismatch of convergence-screen distances

This analysis assumes that only camera convergence distance and screen distance are mismatched (i.e., $k_s = 1$, $k_f = 1$, and $T = [0, 0, 0]^\mathsf{T}$) and camera convergence distance is constant (i.e., $d_c = 3m$). The transfer function (1) can be simplified the ratio of screen distance to camera

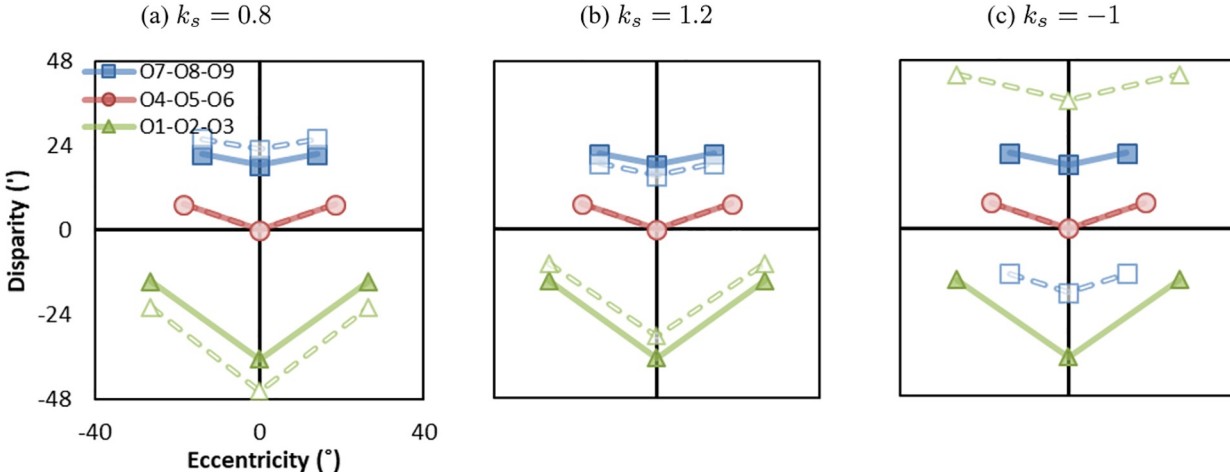

**Fig 7. Effect of the separation mismatches.** (a) Eye separation is smaller than camera separation ($k_s = 0.8$), (b) Eye separation is larger than camera separation ($k_s = 1.2$), (c) Eye separation and camera separation are reversed ($k_s = -1$), called psudoscope. Solid lines represent the eccentricity-disparity in the original world ($\mathcal{E}_{oc}, \mathcal{D}_o$) and dashed lines represent the eccentricity-disparity in the reconstructed world ($\mathcal{E}_{pc}, \mathcal{D}_p$).

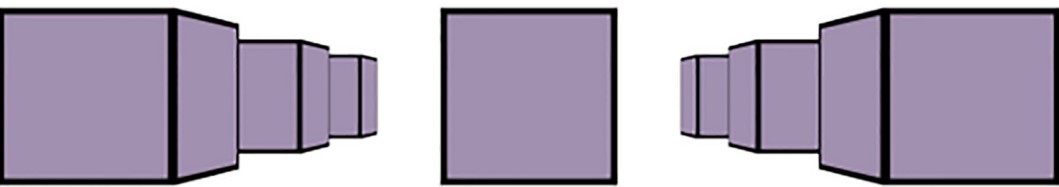

**Fig 8. Effect of distance mismatches from the cyclopean eye (i.e., only monocular depth cues).** The brown cubes are the simulations in the original world and the blue cubes are the simulations in the reconstructed world. They are identical from the view of the cyclopean eye.

convergence distance, $k_d$ as follows:

$$P = \begin{bmatrix} X_p \\ Z_p \end{bmatrix} = \frac{k_d d_c}{Z_o(1 - k_d) + k_d d_c} \begin{bmatrix} X_o \\ Z_o \end{bmatrix}, \tag{9}$$

The visual eccentricities in the original world and reconstructed world are computed by

$$\mathcal{E}_{pc} = \arctan \frac{X_p F_{pz} - Z_p F_{px}}{X_p F_{px} + Z_p F_{pz}} = \arctan \frac{X_o F_{oz} - Z_o F_{ox}}{X_o F_{ox} + Z_o F_{oz}} = \mathcal{E}_{oc}, \tag{10}$$

by substituting $X_p$ and $Z_p$ in (9). Thus, all visual elements share exactly the same eccentricities in conditions of distance mismatches. Fig 8 shows simulations of cubes from the cyclopean eye in the original world (brown cubes) and the reconstructed world (blue cubes) when screen distance mismatches with convergence distance (e.g., $k_d$ = 0.33, a case that a user watches 3D videos on a desktop monitor at distance when convergence distance is $3m$). The same as the mismatch of camera-eye separations, the brown cubes and blue cubes are perfectly matched. Therefore, from monocular perception, the views of the original and the reconstructed worlds are identical.

We consider three different 3D movie screen distance options as examples: $1m$ (desktop monitor viewing distance), $3m$ (TV screen viewing distance), and $10m$ (movie theater screen viewing distance). If convergence distance is set as TV screen viewing distance $3m$ and the 3D movie played on desktop monitor or movie theater screen, the distance ratio $k_d$ is in the range of [0.33, 3] (i.e., $[2^{-1.58}, 2^{1.58}]$ with same logarithmic scale distance from 1).

Fig 9 shows examples with distance mismatches in $\mathcal{E}_c - \mathcal{D}$ coordinates. The solid lines represent the original world ($\mathcal{E}_{oc}, \mathcal{D}_o$) and dashed line represent the reconstructed world ($\mathcal{E}_{pc}, \mathcal{D}_p$) in two conditions: screen distance is smaller than convergence distance $k_d = \frac{1m}{3m} = 2^{-1.58} = 0.33$), and screen distance is larger than convergence distance ($k_d = \frac{9m}{3m} = 2^{1.58} = 3$). From binocular perception, when screen distance is smaller (or larger) than camera convergence distance ($k_d = 0.33/3$), objects appear farther (or closer) in peripheral. Note that, when changing the screen distance, we also change the fixation distance. The eccentricity-disparity structures presented in Fig 9 are depth information relative to the fixation object O5. Again, the eccentricities in the reconstructed world (dashed lines) are identical with the original world (solid lines), representing no geometric distortion from monocular perception.

### 3.3 Mismatch of camera-screen FOVs

The analysis assumes that only screen FOV and camera FOV are mismatched (i.e., $k_s = 1$, $k_d = 1$, and $T = [0, 0, 0]^\mathsf{T}$) and camera convergence distance is constant (i.e., $d_c = 3m$), the transfer

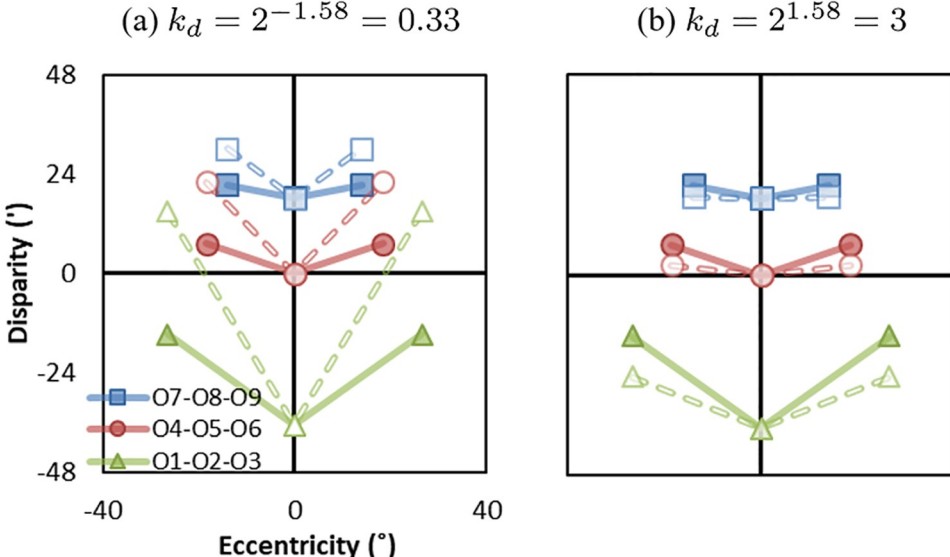

**Fig 9. Effect of distance mismatches.** (a) screen distance is smaller than convergence distance ($k_d = 0.33$), and (b) screen distance is larger than convergence distance ($k_d = 3$). Solid lines represent the eccentricity-disparity in the original world ($\mathcal{E}_{oc}, \mathcal{D}_o$) and dashed lines represent the eccentricity-disparity in the reconstructed world ($\mathcal{E}_{pc}, \mathcal{D}_p$).

function (1) can be simplified with the ratio of screen FOV to camera FOV in linear scale, $k_f$, as follows:

$$P = \begin{bmatrix} X_p \\ Z_p \end{bmatrix} = \frac{k_f d_c}{Z_o(1 - k_f) + k_f d_c} \begin{bmatrix} X_o \\ Z_o/k_f \end{bmatrix}. \tag{11}$$

The visual eccentricities in the reconstructed world can be calculated as

$$\mathcal{E}_{pc} = \arctan \frac{X_p F_{pz} - Z_p F_{px}}{X_p F_{px} + Z_p F_{pz}} = \arctan \frac{X_o F_{oz}/k_f - Z_o F_{ox}/k_f}{X_o F_{ox} + Z_o F_{oz}/k_f^2}. \tag{12}$$

Except the fixation point $F_o = [F_{ox}, F_{oz}]^{\mathsf{T}}$, visual eccentricities in the original world and reconstructed world are different ($\mathcal{E}_{pc} \neq \mathcal{E}_{oc}$). The visual eccentricity in the reconstructed world is the same as the situation when we do not change the sizes (in *xy*-dimension) but extend or compress the depth of objects (in *z*-dimension), i.e.,

$$P = [X_p, Z_p]^{\mathsf{T}} = [X_o, Z_o/k_f]^{\mathsf{T}}, \tag{13}$$

by eliminating the coefficient $\frac{k_f d_c}{Z_o(1-k_f)+k_f d_c}$ in (11). Thus, monocularly, FOV distortions appear depth changes but not size changes.

We consider two conditions of screen FOVs: 90° for Google Cardboard and 37° for desktop monitor (e.g., a 20-inch monitor at $66cm$ distance) when the camera FOV is 60°. Fig 10 shows simulations of cubes from the cyclopean eye in the original world (brown cubes) and the reconstructed world (blue cubes) when FOV ratio is $k_f = \frac{\tan(37°/2)}{\tan(60°/2)} = 2^{-0.79} = 0.58$ (Fig 10a) and FOV ratio is $k_f = \frac{\tan(90°/2)}{\tan(60°/2)} = 2^{0.79} = 1.73$ (Fig 10b). From monocular perception, when screen FOV is smaller than camera FOV ($k_f < 1$), objects appear smaller and farther to the

(a) $k_f = 2^{-0.79} = 0.58$                    (b) $k_f = 2^{0.79} = 1.73$

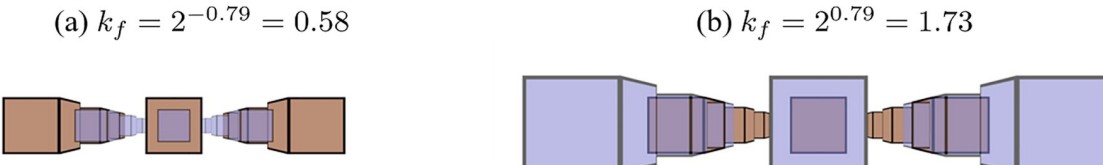

**Fig 10. Effect of FOV mismatch from the cyclopean eye (i.e., only monocular depth cues) in two conditions: (a) screen FOV is smaller than camera FOV ($k_f$ = 0.58), and (b) screen FOV is larger than camera FOV ($k_f$ = 1.73).** The brown cubes are the simulations in the original world and the blue cubes are the simulations in the reconstructed world.

viewer. When screen FOV is smaller than camera FOV ($k_f > 1$), objects appear larger and closer to the viewer.

Fig 11 shows examples of FOV mismatches in $\mathcal{E}_c - \mathcal{D}$ coordinates. The solid lines represent the original world ($\mathcal{E}_{oc}, \mathcal{D}_o$) and dashed lines represent the reconstructed world ($\mathcal{E}_{pc}, \mathcal{D}_p$) in two conditions: screen FOV is smaller than camera FOV ($k_f$ = 0.58) and screen FOV is larger than camera FOV ($k_f$ = 1.73). From binocular perception, when screen FOV is smaller than camera FOV ($k_f$ = 0.58), objects appear smaller in size and compressed in depth towards the screen. When screen FOV is larger than camera FOV ($k_f$ = 1.73), the objects appear larger in size and expended in depth away from the screen. Different from separation mismatches and distance mismatches, the eccentricities in the reconstructed world (dashed lines) decrease or increase compared with that in the original world (solid lines), representing geometric distortions from monocular perception.

## 3.4 Mismatch of head positions

This analysis assumes the orthoscopic reproduction of the scene in S3D ($k_s = 1$, $k_d = 1$, $k_w = 1$), and only the viewer's head translates away from the origin. With these assumptions, the

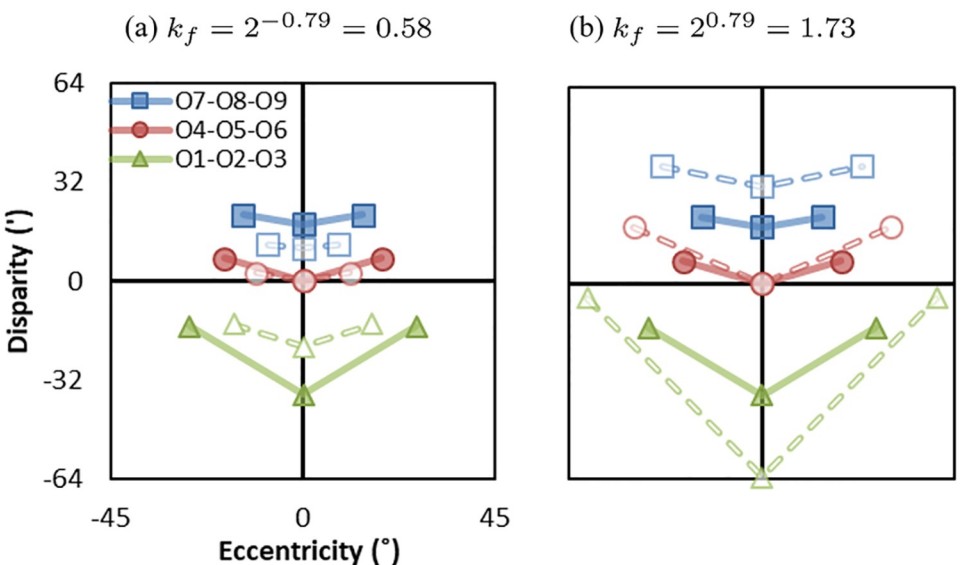

**Fig 11. Effect of FOV mismatch in two conditions: (a) screen FOV is smaller than camera FOV ($k_f$ = 0.58), and (b) screen FOV is larger than camera FOV ($k_f$ = 1.73).** Solid lines represent the eccentricity-disparity in the original world ($\mathcal{E}_{oc}, \mathcal{D}_o$) and dashed lines represent the eccentricity-disparity in the reconstructed world ($\mathcal{E}_{pc}, \mathcal{D}_p$).

(a) Head left ($T_x = -30cm$) (b) Head right ($T_x = 30cm$)

**Fig 12. Effects of head translations horizontally (*x*-axis) from the cyclopean eye (i.e., only monocular depth cues).** (a) Left ($T_x = -30cm$) and (b) right ($T_x = 30cm$). The brown cubes are the simulations in the original world and the blue cubes are the simulations in the reconstructed world. The black arrows show the cubes shear to the left and the right.

transfer function (1) can be simplified with the amount of head translation, *T* as follows,

$$P = \begin{bmatrix} X_p \\ Z_p \end{bmatrix} = \begin{bmatrix} X_o \\ Z_o \end{bmatrix} + \frac{d_c - Z_o}{d_c} \begin{bmatrix} T_x \\ T_z \end{bmatrix}. \tag{14}$$

When the head translates in the amount of *T*, the cyclopean eye is at $E_c = [T_x, T_z]^\top$. The visual eccentricities in the reconstructed world and the original world are different (i.e., $\mathcal{E}_{pc} \neq \mathcal{E}_{oc}$) except for the fixation point $F_o = [F_{ox}, F_{oz}]^\top$. Fig 12 shows simulations of cubes from the cyclopean eye in the original world (brown cubes) and the reconstructed world (blue cubes) when the viewer's head translates to the left $T_x = -30cm$ (Fig 12a) and to the right $T_x = 30cm$ (Fig 12b). From monocular perception, when the viewer's head translates to the left or right, the cubes in front of the screen shear to the left or right (e.g., cube 1 in Fig 12a and 12b) and cubes behind the screen shear to the right or left (e.g., cube 7 in Fig 12a and 12b).

Fig 13 shows simulations of cubes from the cyclopean eye in the original world (brown cubes) and the reconstructed world (blue cubes) when the viewer's head translates to the backward $T_z = -30cm$ (Fig 13a) and to the forward $T_z = 30cm$ (Fig 13b). From monocular perception, when the viewer's head moves backward or forward, the cubes are expanded away from the screen or compressed towards the screen (e.g., cube 1, 7 in Fig 13(a) and 13(b)).

Fig 14 shows examples of head translations in $\mathcal{E}_c - \mathcal{D}$ coordinates. The solid lines represent the original world ($\mathcal{E}_{oc}, \mathcal{D}_o$) in four conditions: the head translates left ($T_x = -30cm$), the head translates right ($T_x = 30cm$), the head translates backward ($T_z = -30cm$), and the head translates forward ($T_z = 30cm$). The dashed lines represent the reconstructed world ($\mathcal{E}_{pc}, \mathcal{D}_p$) in the four conditions. When the head translates left ($T_x = -30cm$), objects in front of the screen are shifted left (eccentricity differences of O1, O2, and O3 are negative) and objects behind the screen are shifted right (eccentricity differences of O7, O8, and O9 are positive). When the

(a) Head backward ($T_z = -30m$) (b) Head forward ($T_z = 30m$)

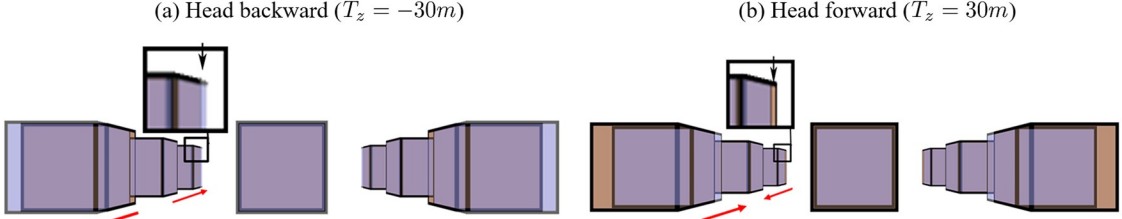

**Fig 13. Effects of head translations along the depth (*z*-axis) from the cyclopean eye (i.e., only monocular depth cues).** (a) Backward ($T_z = -30cm$) and (d) forwards ($T_z = 30cm$). The brown cubes are the simulations in the original world and the blue cubes are the simulations in the reconstructed world. The red arrows show the cubes become closer and farther, representing compression or expansion in S3D space.

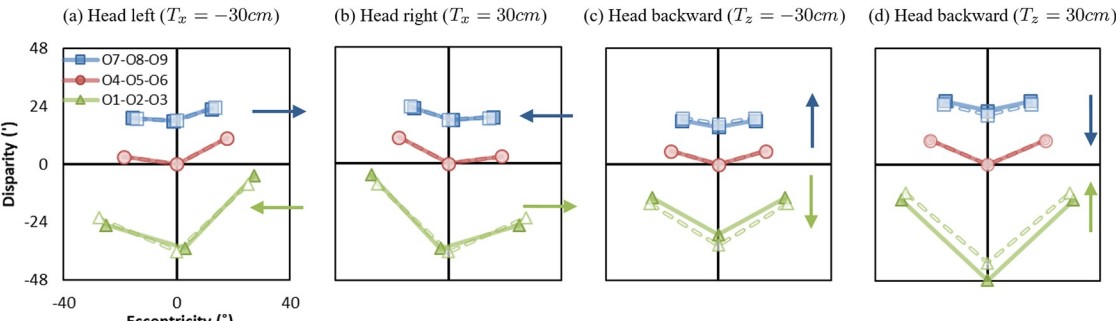

**Fig 14. Effects of head translations in four conditions: The head translates (a) left ($T_x = -30cm$), (b) right ($T_x = 30cm$), (c) backward ($T_z = -30cm$), and (d) forwards ($T_z = 30cm$).** Solid lines represent the eccentricity-disparity in the original world ($\mathcal{E}_{oc}$, $\mathcal{D}_o$) and dashed lines represent the eccentricity-disparity in the reconstructed world ($\mathcal{E}_{pc}$, $\mathcal{D}_p$). Horizontal arrows show eccentricity differences and vertical arrows show disparity differences.

head translates right ($T_x = 30cm$), objects in front of the screen are shifted right (eccentricity differences of O1, O2, and O3 are positive) and objects behind the screen are shifted left (eccentricity differences of O7, O8, and O9 are negative). Note that, Fig 14a and 14b show the same effect as Fig 10 in [3] that has errors and was corrected in [33].

When the head translates backward ($T_z = -30cm$), objects appear compressed toward the screen in depth (disparity differences of O1, O2, and O3 are negative and O7, O8, and O9 are positive). When the head translates forward ($T_z = 30cm$), objects appear expanded away from the screen (disparity differences of O1, O2, and O3 are positive and O7, O8, and O9 are negative). Overall, onscreen objects have the same eccentricities and disparities in the original and reconstructed world. Fixated objects remain at zero eccentricity with head translations. In terms of perception, objects appear to always follow head movements.

## 4 Discussion

The mismatches of camera-eye separations and convergence-screen distances in S3D do not change the monocular depth cues. However, binocularly, these mismatches result in compression or expansion of objects in depth. Pseudoscope on S3D displays is a special case of the mismatch between the camera and the eye separation, where the left and the right view are projected to the right and the left eye, respectively ($k_s = -1$). The binocular disparity depth cue is reversed with respect to the screen while monocular depth cues present veridical depth information. Therefore, these mismatches result in depth-cue conflicts between the monocular and binocular.

The mismatch of camera-screen FOVs, i.e., the screen size is too small or too large, results in scaling of objects in size but without changing the distance of objects on the screen. Monocularly, the depth perception may depend on whether the viewer is familiar with the objects. For familiar objects, minification or magnification of objects increases or decreases the distance judgment, respectively [34, 35]. As a result, object distances estimated from monocular and binocular depth cues are inconsistent. For unfamiliar objects, the viewer does not have any prior of the sizes of objects and may not discern any depth cue conflicts. As discussed in Section '*Distortion-free scaled reproduction*' and '*Correct geometric distortions*' of [4], this provides an approach to eliminate or compensate geometric distortions in S3D by adjusting different parameter pairs, so that the S3D world is only scaled from the original world but without distortions.

Under the conditions when the viewer translates the head away from the origin, i.e., left and right or backward and forward, objects in front of the screen are sheared in the same direction as the head translation and objects behind the screen are sheared in the opposite direction. It appears as if objects follow the movements of the viewer's head. This is because S3D displays can only provide the views captured by the cameras. Thus, the depth cue of motion parallax that exists in real life is missing, which results in a strong perception of object rotation following the viewer's movements [3]. Therefore, the viewer's head translations and the absence of motion parallax conflict.

Visually induced motion sickness (VIMS) involves motions. If the viewer watches a stationary S3D scene and stays still, geometric distortions with depth-cue conflicts may not cause any motion sickness symptoms. In a dynamic scene, for instance, when objects move towards the user in a distorted S3D space where the convergence distance is larger than screen distance (Section 3.2), monocular depth cues remain veridical while the binocular depth cue suggests objects at near or far distances seem to approach the viewer slower or faster than the speed expected. Depth cue conflicts with motions may cause VIMS in S3D, which can be explained by the sensory rearrangement theory [9].

Gao *et al*. [4] proposed a geometric model for S3D and is the most related work. Gao *et al*. [4] analyzed the geometric distortions only based on the binocular depth cue and left a gap between geometric distortions and VIMS in S3D, i.e., the reason why geometric distortions may cause VIMS was not explicitly explained. This work bridges the gap by analyzing depth-cue conflicts and distinguishes from Gao *et al*. [4] in three aspects. First, we introduce a retinal disparity model to analyze geometric distortions in the eccentricity-disparity ($\mathcal{E}_c - \mathcal{D}$) coordinates. The angular disparity ($\mathcal{D}$) represents the binocular depth cue and the visual eccentricity ($\mathcal{E}_c$) represents the monocular depth cues. As demonstrated in the horizontal and the vertical axis of Figs 4, 7, 9, 11 and 14, the geometric distortions in terms of the monocular and the binocular are disentangled and can be discussed separately. Second, we simulate geometric distortions from the cyclopean eye to visually demonstrate the monocular perception, as illustrated in Figs 5, 6, 8, 10, 12 and 13. Third and most importantly, with the help of the retinal disparity model and the visualization technique, the inconsistency between the monocular and binocular depth cues can be clearly analyzed, which bridges the gap between geometric distortions and VIMS in S3D.

In the study of Ichikawa and Egusa [36], subjects with left-right reversing spectacles (i.e., pseudoscope) were all had serious sickness on the first day. Even though depth-cue conflicts in other geometric distortions analyzed above may not be as severe as in pseudoscope, they may result in motion sickness to viewers in a dynamic scene as well. Psychophysical experiments need to be conducted to examine the cause-and-effect of depth-cue conflicts and motion sickness in the future.

Shimojo and Nakajima [37] conducted another pseudoscope experiment where subjects wore left-right reversing spectacles continuously for 9 days. On day 3 of the wearing period, the relation between the direction of the disparity of line-contoured stereograms (LCSs) and the direction of perceived depth was reversed completely. In [36], six subjects wore left-right reversing spectacles continuously for 10 or 11 days. The relation between the direction of physical depth (convex or concave) and the direction of binocular disparity (crossed or uncrossed) was reversed. Also, the subjects' sickness gradually disappeared on about the third day of the wearing period. These studies suggest that binocular disparity depth cue is adaptable to the environment. For 3D producers, e.g., 3D games creators, it may be helpful to present some 3D demonstrations first before the 3D content so that the viewers can gradually get used to the 3D environment and reduce the level of motion sickness.

Hands *et al.* [38] investigated the perception of S3D when viewed from an oblique angle using a canonical-form task in which subjects were asked to report their perception of cubes rendered for perpendicular and oblique viewing. The study showed the lack of difference between S3D and 2D when viewing a familiar object at an oblique viewing angle as large as 20˚. A compensation mechanism could work by recovering the true center of projection (e.g., from cues of the vanishing point and screen slant) and interpreting the oblique retinal images accordingly so that content rendered for a frontoparallel screen appears veridical even when viewed obliquely [39, 40]. Another compensation mechanism could work by the largely unaffected monocular depth cues in a distorted S3D space, as a result of the comparison between [38] and [41]. In [41], objects appeared distorted when viewed obliquely in S3D (i.e., compensation was abolished) since the stimuli of wire-frame hinges in [41] has weaker monocular depth cues compared to the stimuli of texture-solid cubes in [38]. For typical applications of S3D displays in entertainment, S3D content includes a large amount of monocular depth cues and appears relatively veridical when viewed from an oblique angle, which helps explain why S3D content is popular and effective commercially. However, relatively veridical perception does not mean without any issues since people complain about VIMS during or after S3D viewing. Depth-cue conflicts between the monocular and the binocular in a dynamic scene or the absence of motion parallax caused by user-initiated movements could explain the VIMS complaint and the reason why S3D could not spread any further.

In this paper, we assume no viewer's head rotations relative to the screen. This assumption does hold if the viewer sees S3D imagery in head-mounted displays, or the viewer's head stays up relative to the screen. We also assumed that camera image plane (i.e., the image plane perpendicular to the camera axes) and the screen image plane (i.e., the image plane on which the screen is located) are matched. As pointed by [42], when the viewer's head is rotated about a vertical axis relative to the stereo display (yaw rotation), the head is rotated about a forward axis (roll rotation), and stereo images captured by convergence-axis but displayed on a flat screen, additional geometric distortions and vertical disparities are introduced. Note that, the vertical screen disparity and eyestrain caused by the vertical screen disparity [43] and vergence-accommodation conflict [44] are out of the scope of this paper.

## Author Contributions

**Conceptualization:** Zhongpai Gao.

**Data curation:** Zhongpai Gao.

**Formal analysis:** Zhongpai Gao.

**Funding acquisition:** Zhongpai Gao.

**Investigation:** Zhongpai Gao.

**Methodology:** Zhongpai Gao.

**Project administration:** Zhongpai Gao.

**Resources:** Zhongpai Gao.

**Software:** Zhongpai Gao.

**Supervision:** Zhongpai Gao.

**Validation:** Zhongpai Gao, Guangtao Zhai.

**Visualization:** Zhongpai Gao.

**Writing – original draft:** Zhongpai Gao.

**Writing – review & editing:** Zhongpai Gao, Guangtao Zhai, Xiaokang Yang.

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
