## [Decision Letter · Decision Letter 0]

1 Jul 2020

PONE-D-20-04436

Stereoscopic 3D geometric distortions analyzed from the viewer's point of view

PLOS ONE

Dear Dr. Gao,

Thank you for submitting your manuscript to PLOS ONE. After careful consideration, we feel that it has merit but does not fully meet PLOS ONE’s publication criteria as it currently stands. Therefore, we invite you to submit a revised version of the manuscript that addresses the points raised during the review process.

In particular, Reviewer #1 raises concerns that the present work does not sufficiently extend beyond the authors' previous work, and does not provide enough analysis of the model and implications thereof (perhaps through simulations, if not experiments), but rather seems largely an exercise of explaining the math behind such a hypothetical analysis without following through on it. I also agree with the Reviewer's point about the focus on eccentricity at the expense of explicitly dealing with the full variety of monocular depth cues. These concerns must be addressed in order for the manuscript to fulfill publication criteria #1, 3, and 4, listed here: https://journals.plos.org/plosone/s/criteria-for-publication .

We look forward to receiving your revised manuscript.

Kind regards,

Christopher R. Fetsch

Academic Editor

PLOS ONE

Journal Requirements:

Correcting geometric distortions in stereoscopic 3D imaging - https://doi.org/10.1371/journal.pone.0205032

In your revision ensure you cite all your sources (including your own works), and quote or rephrase any duplicated text outside the methods section. Further consideration is dependent on these concerns being addressed.

'This work was supported by the National Natural Science Foundation of China 331 (61901259) and China Postdoctoral Science Foundation (BX2019208).'

'The funders had no role in study design, data collection and analysis, decision to publish, or preparation of the manuscript.'

Reviewers' comments:

Reviewer's Responses to Questions

**Comments to the Author**

1. Is the manuscript technically sound, and do the data support the conclusions?

Reviewer #1: Partly

2. Has the statistical analysis been performed appropriately and rigorously? 

Reviewer #1: N/A

3. Have the authors made all data underlying the findings in their manuscript fully available?

Reviewer #1: Yes

4. Is the manuscript presented in an intelligible fashion and written in standard English?

Reviewer #1: Yes

5. Review Comments to the Author

Reviewer #1: This paper presents an analysis of geometric distortions arising in stereoscopic displays of 3D content (S3D). According to the authors, the aim of the analysis is to identify potential triggers of visually induced motion sickness (VIMS) in S3D. They propose that inconsistencies between binocular (disparity) and monocular depth cues are potential triggers.

While I agree that such an analysis is warranted, in my view, the manuscript does not live up to the aims stated.

The paper essentially consists of the presentation of a mathematical model that aims to reconstruct a 3D scene as seen by the cyclopean eye from images captured by a pair of cameras. No experimental data or statistical analysis are presented. I should point out that the mathematical analysis appears sound although it uses tools that are standard in 3D imaging and that are sufficiently described in the authors’ previous papers.

My main problem is that the work appears unfinished for two reasons.

First, it claims that the model simulates a broad range of monocular depth cues (“linear perspective, interposition (occlusion), object sizes, shades and shadows, texture gradients, accommodation and blur, aerial perspective, etc.”), the only metric derived for their characterization is “eccentricity”. No explanation or justification is provided how this single metric can account for all the monocular cues listed. As it turns out, they in fact measure eccentricity on the retina of the imaginary cyclopean eye.

The second major problem is that the results of the analysis appear trivial at least as far as they are presented in the manuscript. Eventually, the authors do not discuss what is the novelty of their results as compared to their or other authors’ previous work in relation to the stated aims of the study. Indeed, except for a brief derivation of the effects of a simple horizontal head translation, no analysis of dynamic distortions is provided that might occur during scene motion.

6. PLOS authors have the option to publish the peer review history of their article (what does this mean?). If published, this will include your full peer review and any attached files.

Reviewer #1: No

---

## [Author Response · Author response to Decision Letter 0]

24 Aug 2020

Manuscript ID.: PONE-D-20-04436 

Title: Stereoscopic 3D geometric distortions analyzed from the viewer's point of view

We thank the academic editor and reviewers for their constructive comments that helped to greatly improve the contents and presentations of this paper. We have revised the manuscript according to their suggestions.

Point-by-point responses to the comments are listed below.

We hope that the revised version of the manuscript and the responses in this letter are satisfactory. The Highlight changes of the manuscript can be found behind this response. 

(1) AE: The present work does not sufficiently extend beyond the authors' previous work, and does not provide enough analysis of the model and implications thereof (perhaps through simulations, if not experiments), but rather seems largely an exercise of explaining the math behind such a hypothetical analysis without following through on it. Reviewer: The second major problem is that the results of the analysis appear trivial at least as far as they are presented in the manuscript. Eventually, the authors do not discuss what is the novelty of their results as compared to their or other authors’ previous work in relation to the stated aims of the study.

Answer 1: We newly add a discussion about the difference between Gao et al. [4] and this paper in the fourth paragraph of Discussion as follows:

Gao et al. [4] proposed a geometric model for S3D and is the most related work. Gao et al. [4] analyzed the geometric distortions only based on the binocular depth cue and left a gap between geometric distortions and VIMS in S3D, i.e., the reason why geometric distortions may cause VIMS was not explicitly explained. This work bridges the gap by analyzing the depth-cue conflict and distinguishes from Gao et al. [4] in three aspects. First, we introduce a retinal disparity model to analyze geometric distortions in the eccentricity-disparity ( Ec − D ) coordinates. The angular disparity ( D ) represents the binocular depth cue and the visual eccentricity ( Ec ) represents the monocular depth cues. As demonstrated in the horizontal and the vertical axis of Figs 4, 7, 9, 11, 14, the geometric distortions in terms of the monocular and the binocular are disentangled and can be discussed separately. Second, we simulate geometric distortions from the cyclopean eye to visually demonstrate the monocular perception, as illustrated in Figs 5, 6, 8, 10, 12, 13. Third and most importantly, with the help of the retinal disparity model and the visualization technique, we can see the inconsistency between the monocular and binocular depth cues to build a connection between geometric distortions and VIMS in S3D.

(2) It claims that the model simulates a broad range of monocular depth cues (“linear perspective, interposition (occlusion), object sizes, shades and shadows, texture gradients, accommodation and blur, aerial perspective, etc.”), the only metric derived for their characterization is “eccentricity”. No explanation or justification is provided how this single metric can account for all the monocular cues listed. As it turns out, they in fact measure eccentricity on the retina of the imaginary cyclopean eye.

Answer 2: We newly add a figure in Fig 2. As illustrated in the figure, the far and the near objects project to the same retina locations of the cyclopean eye, including all the monocular depth cues (e.g., perspective, occlusion, shade and shadow, and texture gradient). If solely based on the visual eccentricity of the cyclopean eye, the far and the near objects are indistinguishable. Thus, the visual eccentricity (Ec) can account for all the monocular depth cues.

(3) Indeed, except for a brief derivation of the effects of a simple horizontal head translation, no analysis of dynamic distortions is provided that might occur during scene motion.

Answer 3: We newly added paragraph in Discussion as follows:

Visually induced motion sickness (VIMS) involves motions. If the viewer watches a stationary S3D scene and stays still, geometric distortions with depth-cue conflicts may not cause any motion sickness symptoms. In a dynamic scene, for instance, when objects move towards the user in a distorted S3D space where the convergence distance is larger than screen distance (Section 3.2), monocular depth cues remain veridical while the binocular depth cue suggests objects at near or far distances seem to approach the viewer slower or faster than the speed expected. Depth cue conflicts with motions may cause VIMS in S3D, which can be explained by the sensory rearrangement theory [9]

---

## [Decision Letter · Decision Letter 1]

4 Sep 2020

PONE-D-20-04436R1

Stereoscopic 3D geometric distortions analyzed from the viewer's point of view

PLOS ONE

Dear Dr. Gao,

Thank you for submitting your manuscript to PLOS ONE. After careful consideration, we feel that it has merit but does not fully meet PLOS ONE’s publication criteria as it currently stands. Therefore, we invite you to submit a revised version of the manuscript that addresses the points raised during the review process.

The manuscript has not been re-reviewed by the original Reviewer #1, but a new reviewer (#2) has provided additional comments and suggestions, some of which overlap with the original comments and thus imply that these issues have not fully been addressed. Many of the new comments seem addressable with minor textual changes to the manuscript, but others are more substantial (e.g. " Addition of a formula" in comment 4.b) and would ideally need to be re-reviewed by this reviewer. The suggestion under 4.a) to add experimental evidence is also a good one but this would not be required for acceptance. And although it is clear from both reviewers' comments that experts in the field would consider this an 'incremental' contribution, that alone is not sufficient to preclude acceptance according to PLOS ONE criteria.

We look forward to receiving your revised manuscript.

Kind regards,

Christopher R. Fetsch

Academic Editor

PLOS ONE

Reviewers' comments:

Reviewer's Responses to Questions

**Comments to the Author**

1. If the authors have adequately addressed your comments raised in a previous round of review and you feel that this manuscript is now acceptable for publication, you may indicate that here to bypass the “Comments to the Author” section, enter your conflict of interest statement in the “Confidential to Editor” section, and submit your "Accept" recommendation.

Reviewer #2: (No Response)

2. Is the manuscript technically sound, and do the data support the conclusions?

Reviewer #2: Yes

3. Has the statistical analysis been performed appropriately and rigorously? 

Reviewer #2: N/A

4. Have the authors made all data underlying the findings in their manuscript fully available?

Reviewer #2: Yes

5. Is the manuscript presented in an intelligible fashion and written in standard English?

Reviewer #2: Yes

6. Review Comments to the Author

Reviewer #2: The paper provides a model of disparity and spatial eccentricity distortion as a result of mismatch between capture and display parameters such as field of view, stereoscopic convergence, viewing distance and position. The paper further demonstrates effects of such distortions in simple example scenarios.

1) Paper strengths.

a) The paper writing is clear. I have not reviewed the previous revision but based on the edits it has notably improved since then.

b) The figures for individual distortions are clear and serve as a nice illustration of consequences for different mismatch scenarios.

2) The abstract

" In this paper, we analyze the geometric distortions from the viewer’s perspective, so that both monocular and binocular depth cues are considered."

a) The abstract distinguishes between 1st and 3rd person perspective. It is unclear what is meant. Do the authors refer to 1st person and 3rd person games? I assume presence of most pictorial cues -- such as texture gradient or occlusions -- is not removed by change of camera perspective. I assume the authors refer to the fact that they evaluate eccentricity from wrt. cyclopean eye. They should state this clearly already in the abstract.

b) Further, the claim of "considering monocular cues" should be specified more accurately. The reader's impression is that the authors model magnitude and distortion of monocular cues (the authors name linear perspective, occlusion, and shadows in the abstract) yet no such analysis is done.

3) Incremental contributions.

a) The paper is an extension of [4]. It uses the very same model of distortion (Eq. 1) but additionally evaluates projection distortion in spatial coordinates. Since this is a minor technical change I consider it an incremental contribution. The rest of the paper is just evaluation of the model for different input parameters. Most of the effects are fairly obvious (eg. he objects will look smaller on a smaller screen = Fig. 11). The most notable observation seems to the distortion of optical flow in non-orthostereoscopic scenario. However, similar observations has already been done in [3] and more recently in

Hwang, Alex D., and Eli Peli. "Stereoscopic Three-dimensional Optic Flow Distortions Caused by Mismatches Between Image Acquisition and Display Parameters." Journal of Imaging Science and Technology 63, no. 6 (2019): 60412-1.

b) The Figure 14 seems to show the same effect as Figure 5 in [3]. Please clarify it the same model as in [3] could be used to predict results presented here.

4) Perceptual considerations.

a) The model in Eq. 1 is purely geometric and has no connection to the HVS. It is not obvious which of these distortions are noticeable and under which conditions. Not all monocular cues are affected by the image scaling - occlusion, aerial perspective stay completely unchanged, while the effect on texture gradient or shading would require deeper analysis. The authors should clarify which monocular cues are they modeling and evaluate why they think the eccentricity scaling is a good model for them. It seems to me that only "object size" can trivially be linked to the scaling of FOV. However, objects are quite commonly scaled even in 2D video as large objects would not easily fit small screens of phones, yet people do not perceive this as a distortion. Given the aforementioned incremental contribution in the model, adding couple of user studies to verify/quantify perceptual visibility of the predicted effects would significantly solidify the paper.

b) The model predicts absolute distortion of both disparity and eccentricity separately and associates deviation in each of these dimensions with depth distortion and deviation from orthostereoscopic viewing. However, I would argue that with care (scaling multiple parameters at once), one should be able to scale the 3D content such that its smaller version is displayed on a smaller screen while maintaining orthostereoscopic viewing conditions consistent with such hypothetical miniature object. At least for unfamiliar objects, there should be no way for viewer to discern such projection. Addition of a formula that would tie the distortions in both domains together and express the anisotropy related to their mismatch could be another valuable contribution distinguishing this paper from previous work.

5) Minor issues:

- L111: "If solely based on the visual eccentricity of the cyclopean eye, the far and the near objects are indistinguishable. Thus, the visual eccentricity (Ec) represents all the monocular depth cues." - This sentence is unclear. I assume that any projection to any (virtual) eye will contain all the monocular cues. How is the cyclopean eye special?

- It is not clear how discussion on L328 -- 360 ties to the results of this paper. It should probably be moved to related work/background.

6) Summary

The paper demonstrates use of existing model of projection distortion for evaluation of 2D distortion in visual field of view. The manuscript contains intuitive visualizations for each of the various distortion scenarios which could be useful as a quick reference for the way how capture and display mismatch affect the stereoscopic projection. However, given the very incremental contribution I would recommend the authors to supplement experimental validation of the distortion visibility. The main consequence of the work proposed in the abstract is that "The inconsistency of depth cues in a dynamic scene may be a source of visually induced motions sickness." Perhaps testing this hypothesis could provide additional value for the reader and show that the model is useful in predicting this problem.

7. PLOS authors have the option to publish the peer review history of their article (what does this mean?). If published, this will include your full peer review and any attached files.

Reviewer #2: No

---

## [Author Response · Author response to Decision Letter 1]

6 Sep 2020

Manuscript ID.: PONE-D-20-04436 

Title: Stereoscopic 3D geometric distortions analyzed from the viewer's point of view

We thank the academic editor and reviewers for their constructive comments that helped to greatly improve the contents and presentations of this paper. We have revised the manuscript according to their suggestions. Point-by-point responses to the comments are listed below.

We hope that the revised version of the manuscript and the responses in this letter are satisfactory. The Highlight changes of the manuscript can be found behind this response. 

(1) The abstract distinguishes between 1st and 3rd person perspective. It is unclear what is meant. Do the authors refer to 1st person and 3rd person games? I assume presence of most pictorial cues -- such as texture gradient or occlusions -- is not removed by change of camera perspective. I assume the authors refer to the fact that they evaluate eccentricity from wrt. cyclopean eye. They should state this clearly already in the abstract. 

Further, the claim of "considering monocular cues" should be specified more accurately. The reader's impression is that the authors model magnitude and distortion of monocular cues (the authors name linear perspective, occlusion, and shadows in the abstract) yet no such analysis is done.

Answer 1: We modified the abstract as follows:

“In previous work of S3D geometric models, geometric distortions have been analyzed from a third-person perspective based on binocular depth cue (i.e., binocular disparity), where monocular depth cues (e.g., linear perspective, occlusion, and shadows) were not considered.” ->

“In previous works of S3D geometric models, geometric distortions have been analyzed from a third-person perspective based on the binocular depth cue (i.e., binocular disparity). A third-person perspective is different from what the viewer sees since monocular depth cues (e.g., linear perspective, occlusion, and shadows) from different perspectives are different. “

(2) The paper is an extension of [4]. It uses the very same model of distortion (Eq. 1) but additionally evaluates projection distortion in spatial coordinates. Since this is a minor technical change I consider it an incremental contribution. The rest of the paper is just evaluation of the model for different input parameters. Most of the effects are fairly obvious (eg. the objects will look smaller on a smaller screen = Fig. 11). The most notable observation seems to the distortion of optical flow in non-orthostereoscopic scenario. However, similar observations has already been done in [3] and more recently in

Hwang, Alex D., and Eli Peli. "Stereoscopic Three-dimensional Optic Flow Distortions Caused by Mismatches Between Image Acquisition and Display Parameters." Journal of Imaging Science and Technology 63, no. 6 (2019): 60412-1.

Answer 2: This paper distinguishes from Gao et al. [4] as discussed in Discussion:

Gao et al. [4] proposed a geometric model for S3D and is the most related work. Gao et al. [4] analyzed the geometric distortions only based on the binocular depth cue and left a gap between geometric distortions and VIMS in S3D, i.e., the reason why geometric distortions may cause VIMS was not explicitly explained. This work bridges the gap by analyzing the depth-cue conflict and distinguishes from Gao et al. [4] in three aspects. First, we introduce a retinal disparity model to analyze geometric distortions in the eccentricity-disparity ( Ec − D ) coordinates. The angular disparity ( D ) represents the binocular depth cue and the visual eccentricity ( Ec ) represents the monocular depth cues. As demonstrated in the horizontal and the vertical axis of Figs 4, 7, 9, 11, 14, the geometric distortions in terms of the monocular and the binocular are disentangled and can be discussed separately. Second, we simulate geometric distortions from the cyclopean eye to visually demonstrate the monocular perception, as illustrated in Figs 5, 6, 8, 10, 12, 13. Third and most importantly, with the help of the retinal disparity model and the visualization technique, the inconsistency between the monocular and binocular depth cues can be clearly analyzed, which bridges the gap between geometric distortions and VIMS in S3D..

Note that, for the mismatch of camera-eye separations and the mismatch of convergence-screen distances (Section 3.1 and 3.2), monocular depth cues are not changed, which is not very obvious if without the help of the retinal disparity model and the visualization tool in Fig 6 and 8. 

We add the citation of Hwang and Peli (2019) that focuses on optic flow distortions while this paper focuses on depth cue conflicts. 

(3) Figure 14 seems to show the same effect as Figure 5 in [3]. Please clarify it the same model as in [3] could be used to predict results presented here.

Answer 3: We newly add a sentence in Section 3.4 as follows 

Note that, Fig 14a and 14b show the same effect as Figure 10 in [3] that has errors and was corrected in [33].

(4) The model in Eq. 1 is purely geometric and has no connection to the HVS. It is not obvious which of these distortions are noticeable and under which conditions. Not all monocular cues are affected by the image scaling - occlusion, aerial perspective stay completely unchanged, while the effect on texture gradient or shading would require deeper analysis. The authors should clarify which monocular cues are they modeling and evaluate why they think the eccentricity scaling is a good model for them. It seems to me that only "object size" can trivially be linked to the scaling of FOV. However, objects are quite commonly scaled even in 2D video as large objects would not easily fit small screens of phones, yet people do not perceive this as a distortion. Given the aforementioned incremental contribution in the model, adding couple of user studies to verify/quantify perceptual visibility of the predicted effects would significantly solidify the paper.

Answer 4: The geometric model in Eq. 1 is derived from the binocular disparity, which is one of the most important binocular depth cues for HVS. Depth perception in a 3D space involves both monocular and binocular depth cues and HVS interprets depth by integrating various depth cues [13–15]. We agree that geometric distortions predicted by the geometric model may not be the same as what the viewer perceives since monocular depth cues are not considered for the geometric model. That is why we introduce the retinal disparity model to analyze geometric models from the viewer’s perspective. 

As illustrated in Fig 2, the far and the near objects project to the same retina locations of the cyclopean eye, including all the monocular depth cues (e.g., perspective, occlusion, shade and shadow, and texture gradient). If solely based on the visual eccentricity of the cyclopean eye, the far and the near objects are indistinguishable. Thus, the visual eccentricity (Ec) can account for all the monocular depth cues. We are not trying to model each monocular depth cue but the visual eccentricity of the cyclopean eye can include all the monocular depth cues. 

We agree that object scaling itself may not be perceived as distortions. We newly add discussion in the second paragraph of Discussion. Importantly, in this paper, we mainly focus on depth cue conflicts, i.e., the inconsistency between the monocular and the binocular depth cues. 

The main contributions of this paper can be summarized as follows:

1) We introduce a retinal disparity model to analyze geometric distortions in the eccentricity-disparity ( Ec − D ) coordinates. The angular disparity ( D ) represents the binocular depth cue and the visual eccentricity ( Ec ) represents the monocular depth cues. As demonstrated in the horizontal and the vertical axis of Figs 4, 7, 9, 11, 14, the geometric distortions in terms of the monocular and the binocular are disentangled and can be discussed separately.

2) We simulate geometric distortions from the cyclopean eye to visually demonstrate the monocular perception, as illustrated in Figs 5, 6, 8, 10, 12, 13. 

3) With the help of the retinal disparity model and the visualization technique, the inconsistency between the monocular and binocular depth cues can be clearly analyzed, which bridges the gap between geometric distortions and VIMS in S3D.

User studies are necessary to verify the hypothesis that depth cue conflicts in a dynamic scene may cause VIMS. Considering the length of this paper and what this paper mainly focusses on, user study is not covered by this paper and will be conducted in the future.

(5) The model predicts absolute distortion of both disparity and eccentricity separately and associates deviation in each of these dimensions with depth distortion and deviation from orthostereoscopic viewing. However, I would argue that with care (scaling multiple parameters at once), one should be able to scale the 3D content such that its smaller version is displayed on a smaller screen while maintaining orthostereoscopic viewing conditions consistent with such hypothetcal miniature object. At least for unfamiliar objects, there should be no way for viewer to discern such projection. Addition of a formula that would tie the distortions in both domains together and express the anisotropy related to their mismatch could be another valuable contribution distinguishing this paper from previous work.

Answer 5: We appreciate this discovery by the reviewer. We revised the Discussion as follows:

The mismatch of camera-screen FOVs, i.e., the screen size is too small or too large, results in scaling of objects in size but without changing the distance of objects on the screen. Monocularly, the depth perception may depend on whether the viewer is familiar with the objects. For familiar objects, minification or magnification of objects increases or decreases the distance judgment, respectively [34] [35]. As a result, object distances estimated from monocular and binocular depth cues are inconsistent. For unfamiliar objects, the viewer does not have any prior of the sizes of objects and may not discern any depth cue conflicts. As discussed in Section `Distortion-free scaled reproduction' and `Correct geometric distortions' of [4], this provides an approach to eliminate or compensate geometric distortions in S3D by adjusting different parameter pairs, so that the S3D world is only scaled from the original world but without distortions. 

Note that, the compensation between different parameter mismatches has been discussed in the section of “Correct geometric distortions” in Gao et al. [4]. This paper is not trying to correct or remove geometric distortions but to point out that depth cue conflicts in a dynamic scene may be a source of VIMS. 

(6) L111: "If solely based on the visual eccentricity of the cyclopean eye, the far and the near objects are indistinguishable. Thus, the visual eccentricity (Ec) represents all the monocular depth cues." - This sentence is unclear. I assume that any projection to any (virtual) eye will contain all the monocular cues. How is the cyclopean eye special?

Answer 6: We discuss the cyclopean eye because it can represent the monocular view of the viewer. Discussing any other virtual eye is meaningless since it is not related to what the viewer sees. For binocular vision, the far and the near objects are distinguishable because of the binocular disparity. 

(7) It is not clear how discussion on L328 -- 360 ties to the results of this paper. It should probably be moved to related work/background.

Answer 7: We would like to clarify that L328 – 360 are in Discussion not in Results. In the paragraph of “Shimojo and Nakajima [37] …”, they conducted a pseudoscope experiment, which is one of the cases in this paper. This study suggests that

The binocular disparity depth cue is adaptable to the environment. For 3D producers, e.g., 3D games creators, it may be helpful to present some 3D demonstrations first before the 3D content so that the viewers can gradually get used to the 3D environment and reduce the level of motion sickness.

In the paragraph of “Hands et al. [38]…”, we discuss that

For typical applications of S3D displays in entertainment, S3D content includes a large amount of monocular depth cues and appears relatively veridical when viewed from an oblique angle, which helps explain why S3D content is popular and effective commercially. However, relatively veridical perception does not mean without any issues since people complain about VIMS during or after S3D viewing. Depth-cue conflicts between the monocular and the binocular in a dynamic scene or the absence of motion parallax caused by user-initiated movements could explain the VIMS complaint and the reason why S3D could not spread any further.

The reason that we put them in Discussion is that we are trying to discuss the connection between their results and this paper. We believe it is clearer to discuss them after introducing our work. 

(8) The paper demonstrates use of existing model of projection distortion for evaluation of 2D distortion in visual field of view. The manuscript contains intuitive visualizations for each of the various distortion scenarios which could be useful as a quick reference for the way how capture and display mismatch affect the stereoscopic projection. However, given the very incremental contribution I would recommend the authors to supplement experimental validation of the distortion visibility. The main consequence of the work proposed in the abstract is that "The inconsistency of depth cues in a dynamic scene may be a source of visually induced motions sickness." Perhaps testing this hypothesis could provide additional value for the reader and show that the model is useful in predicting this problem.

Answer 8: We thank the reviewer for pointing out the necessity to conduct psychophysics experiments to test our hypothesis. We believe psychophysics experiments should be put in an independent paper considering the length of the paper and what this paper mainly focuses on.

---

## [Editor Report · Decision Letter 2]

1 Oct 2020

Stereoscopic 3D geometric distortions analyzed from the viewer's point of view

PONE-D-20-04436R2

Dear Dr. Gao,

We’re pleased to inform you that your manuscript has been judged scientifically suitable for publication and will be formally accepted for publication once it meets all outstanding technical requirements.

Kind regards,

Christopher R. Fetsch

Academic Editor

PLOS ONE
---

## [Editor Report · Acceptance letter]

7 Oct 2020

PONE-D-20-04436R2 

Stereoscopic 3D geometric distortions analyzed from the viewer’s point of view 

Dear Dr. Gao:

I'm pleased to inform you that your manuscript has been deemed suitable for publication in PLOS ONE. Congratulations! Your manuscript is now with our production department. 

Kind regards, 

on behalf of

Dr. Christopher R. Fetsch 

Academic Editor

PLOS ONE